# Cyclodextrin Inclusion Complexes and Their Application in Food Safety Analysis: Recent Developments and Future Prospects

**DOI:** 10.3390/foods11233871

**Published:** 2022-11-30

**Authors:** Jiaojiao Zhou, Jilai Jia, Jiangling He, Jinjie Li, Jie Cai

**Affiliations:** 1National R&D Center for Se-Rich Agricultural Products Processing, Hubei Engineering Research Center for Deep Processing of Green Se-Rich Agricultural Products, School of Modern Industry for Selenium Science and Engineering, Wuhan Polytechnic University, Wuhan 430023, China; 2Institute of System and Engineering, Beijing 100010, China; 3Key Laboratory for Deep Processing of Major Grain and Oil, Ministry of Education, Hubei Key Laboratory for Processing and Transformation of Agricultural Products, Wuhan Polytechnic University, Wuhan 430023, China

**Keywords:** cyclodextrins, cyclodextrin inclusion complexes, food safety

## Abstract

Food safety issues are a major threat to public health and have attracted much attention. Therefore, exploring accurate, efficient, sensitive, and economical detection methods is necessary to ensure consumers’ health. In this regard, cyclodextrins (CDs) are promising candidates because they are nontoxic and noncaloric. The main body of CDs is a ring structure with hydrophobic cavity and hydrophilic exterior wall. Due to the above characteristics, CDs can encapsulate small guest molecules into their cavities, enhance their stability, avoid agglomeration and oxidation, and, at the same time, interact through hydrogen bonding and electrostatic interactions. Additionally, they can selectively capture the target molecules to be detected and improve the sensitivity of food detection. This review highlights recent advances in CD inclusion technology in food safety analysis, covering various applications from small molecule and heavy metal sensing to amino acid and microbial sensing. Finally, challenges and prospects for CDs and their derivatives are presented. The current review can provide a reference and guidance for current research on CDs in the food industry and may inspire breakthroughs in this field.

## 1. Introduction

Cyclodextrins (CDs) are water-soluble, nontoxic, and nonreducing cyclic oligosaccharides made up of D-glucopyranoside units linked by α-1,4-glycosidic bonds [1]. The most common CDs are α-CD, β-CD, and γ-CD, composed of 6, 7, and 8 glucose subunits with diameters of 0.5, 0.6, and 0.8 nm, respectively. The positions of hydroxyl groups in CDs endow them with a hydrophilic property at the rims. The interior cavity of CDs is hydrophobic because of the glycoside-bonds orientation [2]. This unique property enables CDs to form inclusion complexes with hydrophobic guest molecules [3,4]. CDs are widely used in biodetection, biomedicine, bioimaging, agricultural production, and so forth [5]. During the past decades, benefiting from the development of material science, significant advances have been achieved in developing high-performance CDs, regulating the CD structure, and broadening potential applications (Figure 1).

A Web of Science search demonstrates that about 30000 CD journal articles have been reported. Accompanying the rapid development of CD studies, some excellent CD reviews of food science have appeared [6,7,8,9,10]. However, most reviews focus on food nutrition, functionality, and delivery. Furthermore, few reviews have specifically reported the applications of novel CD-based systems in food safety analysis. Therefore, this review highlights the recent developments in CD inclusion technology in the food safety analysis (Figure 1). In this review, we first introduce the characteristics and classifications of CDs. Subsequently, we summarize various strategies for preparing CD inclusion complexes. Next, the broad applications from small molecules and heavy metal sensing to microorganism sensing are covered. Finally, the challenges and future perspectives of CDs and their derivatives are proposed. This review can provide a reference on the study of CD in food industry and may stimulate breakthroughs in this field.

## 2. Structure and Physicochemical Properties of Cyclodextrins

As mentioned above, the commonly used CDs are the natural α, β, and γ-forms. Moreover, CDs with more than eight glucose units or less than six glucose units have also been reported. Their physical and chemical properties differ from each other because of their different structures, as shown in Table 1.

### 2.1. α-Cyclodextrin

α-CD consists of six glucose units with the molecular formula C_36_H_60_O_30_ and a molecular weight of 972.84 Da. A small lumen and high resistance to enzymatic hydrolysis endow α-CD with broad applications in many fields, particularly in the food industry. Although CDs are water-soluble, their solubility is different [11]. α-CD has modest solubility in water that is almost multifold higher than that of β-CD but approximately 1.6 times lower than that of γ-CD at 25 °C [12,13].

The three CDs are thermally stable (above 200℃) and stable in alkaline or moderately acidic solutions [13]. Compared with β-CD and γ-CD, α-CD is considerably more resistant to hydrolysis in acid solutions. The three CDs are stable in the presence of glucoamylase or β-amylase, but they can be hydrolyzed by some α-amylases. Additionally, α-CD is Generally Regarded As Safe (GRAS) by the Food and Drug Administration (FDA). However, because of its small molecular cavity, α-CD can only encapsulate small molecules, which limits its applications. The applications of α-CD are also related to its low digestibility and some indigestion risk for large amounts. Different chemical derivatives of α-CD have been prepared by modifying its hydroxyl groups [14,15]. For example, cyclodextrin glycosyltransferase catalyzes the enzymatic conversion of starch or starch derivatives to α-CD. However, because of its low yield and high cost, its market share is far less than that of β-CD.

Significant work has been conducted to improve the production process of α-CD by altering the properties of α-CD glycosyltransferase. Advances in biotechnology are expected to significantly improve the production process of high-purity α-CD and expand its application in the food industry [13].

### 2.2. β-Cyclodextrin

β-CD comprises seven glucose units, the most used CDs with a molecular formula of C_42_H_70_O_35_ and a molecular weight of 1135 Da. This is because the medium cavity size of β-CD is more stable, allowing a complex of various guest molecules with a high yield and an acceptable price. The flexible hydroxyl groups around the narrower edges can form hydrogen bonds [13], resulting in the highest hydrogen bond strength of β-CD and the lowest solubility among the three CDs.

β-CD derivatives can be prepared via introducing functional groups while keeping the composition of CD unchanged [9]. Researchers have developed different chemically derived derivatives (for example, hydroxypropyl-β-CD) and materials containing β-CD (for example, CD-based nanoparticles) [6]. Both β-CD and hydroxypropyl-β-CD have the GRAS status from the FDA [16]. 

### 2.3. γ-Cyclodextrin

The molecular formula of γ-CD is C_48_H_80_O_40_, and its molecular weight is 1297 Da. It was first discovered in 1935 and is the earliest discovered among the three natural CDs [17].

**Table 1 foods-11-03871-t001:** Physicochemical properties of the three natural CDs.

Physicochemical Properties	α-CD	β-CD	γ-CD	References
Glucose unit	6	7	8	[1]
Chemical formula	C_36_H_60_O_30_	C_42_H_70_O_35_	C_48_H_80_O_40_	[9]
Molecular weight (Da)	972	1135	1297	[18]
Diameter of central cavity (nm)	0.57	0.78	0.95	[10]
Outer diameter (nm)	1.4–1.5	1.5–1.6	1.7–1.8	[9]
Melting point (°C)	275	280	275	[9]
pKa at 25 °C	12.3	12.2	12.1	[9]
Internal water molecules	6–8	11–12	13–17	[9]
Solubility in water at 25 °C (mg/mL)	145	18.5	232	[18]

Among the three CDs, γ-CD has the largest cavity and the highest solubility and safety [18]. Unfortunately, the high cost of enzyme preparation and purification process limit the large-scale production of γ-CD [19]. The lower yield and superior performance of γ-CD lead to its high price, which motivates researchers to optimize the preparation process of γ-CD. For example, Wu et al. used an enzyme-assisted pretreatment strategy to improve the yield of γ-CD under swelling conditions. As a result, the yield increased from 10.43% to 26.34%, significantly higher than that of the previous report [20].

### 2.4. Other Structures

Apart from three natural CDs, CDs with nineteen units have also been characterized [21,22]. However, these novel CDs are not commonly used because of their instability. Furthermore, when the number of glucose units is less than six, CDs cannot be cyclized because of their steric overlap. In this regard, Daiki et al. proposed a new strategy for the synthesis of CDs with three or four glucose units, whose smaller cavities may allow for the selective inclusion of molecules more minor than those accommodated in currently available CDs while improving stability [23].

## 3. Inclusion Complex Formation in CDs

The upper and lower rims of native CDs are hydrophilic, and the inner cavity is hydrophobic, which can act as a host to entrap various appropriate guests. When guest molecules enter the inner cavity of CDs, an “inclusion complex” is formed. These coating CD molecules can improve the guests’ chemical, physical and biological properties, including stability and bioactivity [24]. The properties of guest molecules can influence the complexation or dissociation of the CD inclusion complex. The stability of the inclusion complex can be described by the stability constant (Ks) [9]:(1)CD+G=CD−G
(2)$$\;\mathrm{Ks}=\mathrm{Kr}/\mathrm{Kd}=\left[\mathrm{CD}-G\right]/\left[\mathrm{CD}\right]\left[G\right]$$
where G is the guest molecule, and CD − G is the inclusion complex. The stability of the inclusion complex is calculated by the stability constant Ks. The larger the ratio is, the easier the inclusion complex is to form and more stable [25].

The structure and properties of the inclusion complex could be characterized and evaluated. The commonly used methods include thermal analysis [26], nuclear magnetic resonance (NMR) [27], X-ray diffraction (XRD) [28], Fourier transform infrared spectroscopy (FTIR) [29], atomic force microscope (AFM) [30], and so on. Recently, cheminformatics techniques, such as molecular docking and dynamics, have gained increasing popularity because of their increased ability to predict embedded complexes [31,32,33,34]. The interfaces of CDs have been used for input and output analysis, and there is even a library of computational molecular dynamics simulations for CDs [35]. Specifically, molecular docking can be used to predict the preferred orientation of a host toward a guest. Conversely, the knowledge of preferred orientation can be used to predict the binding affinity between two molecules [36,37]. The formation of the inclusion complex is one of the most exceptional characteristics of CDs. Truncation of CDs results in a series of unique complexes between CDs and many other molecules, thus forming a polymer called host–guest or inclusion complex. 

Barrel-shaped CDs molecules trap guest molecules in their cavities to form inclusion complexes through hydrogen-bonding interactions. The lipophilic cavity of CDs provides a suitable environment for hydrophobic guest molecules. There is no particular need for any covalent bonds since electrostatic interactions or van der Waals forces bind when the host reaches the cavity. Therefore, covalent bonds will not be formed or broken during packaging, and only specific and suitable compounds can form complexes with CDs [38]. In general terms, hydrophobic molecules form complexes with CDs, although neutral or polar molecules, ions or gases are also able to form such complexes. The solvents are important since the presence of organic solvents might decrease the complexation constant due to the higher solubility of the molecule [39]. On the other hand, the presence of a minimal quantity of water is necessary to form the inclusion complex. 

The entrapment of different molecules has the advantages of stabilizing compounds, preventing the oxidation of various components, controlling sublimation, improving sensory quality, and preventing peculiar smells, among others. It also helps to physically separate incompatible compounds, stabilize the flavor, enhance the antibacterial activity, and promote intelligent packaging. Therefore, it is not sufficient to investigate the formation mechanism of the CD inclusion complex. Therefore, further efforts need to be devoted to analyzing the metabolic mechanisms of CD inclusion complex and their effects on human health.

## 4. Preparation Method of Cyclodextrins Inclusion Complex

Several methods for preparing CD inclusion complexes exist; however, no universal methods exist. Therefore, it is necessary to choose the appropriate method to prepare CD inclusion complexes according to the properties of the guest molecules [40]. The commonly used methods for preparing CD inclusion complexes include the saturated aqueous solution method, physical mixing method, spray drying method, freeze–drying method, colloid grinding method, microwave irradiation method, supercritical fluid method, and so on. A comparison of the various methods for the preparation of CD inclusion complex is list in Table 2. 

### 4.1. Saturated Aqueous Solution Method

The saturated aqueous solution method, also known as the precipitation method, is suitable for substances that are not soluble in water. It has the advantages of simplicity and high efficiency. The guest molecules are added to the saturated CDs aqueous solution, and the mixture is stirred for a specific time at an appropriate temperature, followed by cooling, crystallization, and drying. As a result, the inclusion compound is obtained. For example, Jiang dissolved β-CD in an aqueous ethanol solution, followed by the addition of catechin (CAT) under stirring. The CAT/β-CD inclusion complex was obtained by vacuum filtration after cryoprecipitate. Consequently, CAT was effectively protected by encapsulation into β-CD, and its antioxidant stability was significantly improved [41]. Similarly, Li et al. incorporated allyl isothiocyanate into the hydrophobic cavities of α-CD and β-CD. Using the co-precipitation method combined with water washing, the inclusion rate can reach almost 100% [42]. Additionally, some drug-CDs complexes can be precipitated by adding antisolvents [43].

Although the saturated aqueous solution method has become the commonly used methods for preparing inclusion complexes due to its simplicity and high entrapment efficiency [41], large-scale production produces so much wastewater. Additionally, the subsequent wastewater treatment process is very complex, and the large-scale production technology and equipment are not yet mature. Therefore, this method is usually used in the laboratory.

### 4.2. Kneading Method

The kneading method (also called paste method) is a moderately simple method, which is suitable for poorly water-soluble guests and includes the addition of dissolved solid guest to a slurry of CD to form a paste in a mortar [8]. A paste is obtained by mixing CDs and water in a vortex mixer, and then guest substances are introduced [10]. The guests can be also complexed by simply adding to the CD and mixing to form a physical mixture. Finally, the mixture is washed, filtered, dried, ground, and sieved. The solvent and the time of kneading depend on the guest. Although the amount of solid clathrate obtained using this method is small, it is simple and efficient, with the potential for large-scale production.

### 4.3. Spray Drying Method

Spray drying is one of the oldest methods for preparing CD inclusion complexes [44]. The method consists of three steps: (1) Atomization of the liquid feed. After dissolving the guest substance, add it to the saturated solution of CDs, set the instrument’s feeding speed, rotation speed, inlet temperature, and outlet temperature, and atomize the mixed liquid under a certain pressure. (2) Mixing and drying. The atomized fine droplets are mixed with a heated air stream for drying. (3) Collect particles. After drying, the products are separated from the gas stream and collected in a sample collection bottle to obtain the powder [44]. This method is unsuitable for highly volatile and heat-labile guest substances and is limited to water-dispersible or water-soluble materials. Additionally, the feed rate should be controlled during spray drying to avoid clogging the nozzles with too large particles [45].

The spray drying method has the advantages of simplicity, high drying speed, and easy large-scale production [9]. However, this method requires high energy consumption and tedious sample preparation for professional instruments. Consequently, it has not been widely used.

### 4.4. Freeze–Drying Method

The freeze–drying method is to dissolve the guest molecule in water (if it is a weakly acidic compound, an appropriate amount of ammonia water can be used to dissolve it according to the actual situation), add CDs in proportion to stirring thoroughly, and freeze–dry the mixed solution to obtain the inclusion compound [46].

The CDs are first dissolved to prepare a saturated solution. Subsequently, the dissolved solution of the guest substance is added into the saturated solution under stirring until equilibrium is reached.

Next, the mixed solution is placed in a freeze dryer, and the freezing time and temperature are set and then dried to obtain the desired power [47]. This method suits guest substances soluble in water or readily decomposed under heating and drying conditions. Under the premise of the need to obtain a powdered inclusion compound, the solvent can also be removed by a freeze–drying method to obtain a powdered inclusion compound.

Freeze-drying is one of the most widespread techniques for improving the stability of compounds. It is conducted at low temperatures, and the water is directly sublimated, so the dried inclusion compound is stable and convenient for long-term storage. Additionally, since the drying of the compound is completed in a frozen state, the material’s physical structure and molecular structure change very little, and its organizational structure and appearance are well preserved. However, the freeze–drying method also has some disadvantages. Most compounds are susceptible to stress during the freezing and drying cycles. Compounds sensitive to temperature and pressure result in reduced process efficiency and poor product quality. However, this method is expensive; therefore, its commercial applicability is severely limited by long processing times [48].

### 4.5. Colloid Grinding Method

The colloid grinding method refers to adding an appropriate amount of deionized water to the CDs, grinding it uniformly, subsequently adding the guest molecules, mixing and grinding, drying at low temperature, washing with an appropriate solvent, and drying to obtain the inclusion compound. Finally, the mixture of guest and CD is subjected to mechanical grinding. When the compound is trapped between the grinding media, the compound can acquire sufficient strength, accumulating quasi-adiabatic energy and, thus, forming a metastable structure [49]. This process causes crystal breakage, particle size reduction, and increased contact surface for guest species to interact with CDs. Generally, the colloid grinding method is especially suitable for poorly soluble guest substances and has obvious advantages such as saving, simplicity, high efficiency, and high quality [50]. Among the different techniques proposed for preparing CD inclusion complexes in the solid state, grinding appears as a fast, convenient, sustainable, and solvent-free method [51,52,53]. Alternatively, the disadvantage of the grinding method is that the production speed is slow, the yield is low, and the strength quickly affects the inclusion rate of the inclusion compound.

### 4.6. Supercritical Fluid Method

The supercritical fluid method is a new method for preparing CDs inclusion compounds, which mainly uses the solvent properties of supercritical fluids at higher than critical temperatures and pressure to prepare inclusion compounds, carbon dioxide (CO_2_) is usually used as the supercritical fluid [54,55].

CO_2_ is first flushed into a high-pressure vessel containing the guest substance and CDs and stirred at an appropriate temperature. Then, the mixture is pumped into a predetermined pressure system for a fixed period. The procedure is terminated by rapid decompression to atmospheric pressure, which involves CO_2_ gasification and its separation from the obtained complex [56]. The supercritical fluid method can avoid using organic solvents and effectively solve the problem of solvent residues. This method has been employed to introduce a significant quantity of guest molecules into the cavities of CDs. Since steps, including solvent removal, are reduced, the preparation process of the inclusion complex is also greatly simplified. Additionally, this method decreases the inclusion temperature and is suitable for substances sensitive to moisture and heat [57]. 

In summary, different techniques to prepare CD inclusion complexes have been introduced. Among these methods, saturated aqueous solution, physical mixing, spray drying, and freeze drying have been the most used techniques in the last few years. CD inclusion complexes have been widely used in food packaging in the last decades. Nowadays, CD inclusion complexes are useful to obtain edible films and coating food with active properties. 

## 5. Application of CDs and Their Derivatives in the Food Safety Analysis

CDs have been widely used in the food industry because of their low cost, availability, and nontoxicity. On the one hand, CDs can capture contaminants because of its unique structural properties, making it suitable as an adsorbent [58]. Alternatively, CDs can easily accommodate various suitable small molecules to prepare functional composites for biosensing. Consequently, these features endow CDs with broad applications in food safety analysis. Although CDs have received extensive attention, the summary of their application in food safety is still not thorough. In this section, we summarize the latest applications of CDs in food safety analysis according to the types of targets.

### 5.1. Additives

Food additives are non-nutritive substances intentionally added to food in small amounts to improve the food’s quality. For example, CD polymers are used as food additives to enhance antibacterial activity and shelf life [59], reduce cholesterol content [60], improve the sensory quality of food [61], maintain color, and so forth [62]. CD can also be used to detect illegal additives.

Various pigments endow food with pleasing color, significantly increasing their preference for food and stimulating appetite. Lai et al. prepared native β-CD capped silver nanoparticles (AgNPs) decorated Ti_3_C_2_T_x_ nanosheets (Ti_3_C_2_T_x_-AgNPs@β-CD) surface-enhanced Raman scattering (SERS) substrates. The two-dimensional Ti_3_C_2_T_x_ nanosheets not only acted as the supporter to load AgNPs but also served as chemical enhancement material. Meanwhile, β-CD played a critical role in the control of morphology and size of AgNPs, and its supramolecular structure also endowed the Ti_3_C_2_T_x_-AgNPs@β-CD substrate with selectivity. Then, the substrate was applied for the selective detection of erythrosin B among pigment additives via the host-guest size-matching effect [63]. There was a good linear relationship ranged from 1 to 100 mg/L for Erythrosin B assay. Huang et al. reported a robust fluorescent sulfur quantum dots (SQDs) material that possess both molecular recognition and fluorescence detection capability by employing hydroxypropyl-β-CD (HP-β-CD) as ligand (Figure 2). Owing to the synergistic effect of inner filter effect and host-guest interaction, the SQDs were used as an effective fluorescent probe for tartrazine detection with excellent sensitivity and selectivity. There was a wide linear range from 0 to 30 μM with a detection limit of 82 nM for the detection of tartrazine [64]. Furthermore, the fluorescent SQDs were successfully utilized to detect tartrazine in river water and food samples with satisfactory performance.

Tert-butylhydroquinone (TBHQ) is a general synthetic lipid antioxidant widely used as a food preservative. Because it helps scavenge and neutralize free radicals, it prevents food spoilage and rancidity. TBHQ has a relatively good heat resistance and does not alter its odor, taste, or color compared with other antioxidants. However, high concentrations of TBHQ have various dangerous side effects, and excessive intake of TBHQ can lead to conditions including immune system damage, convulsions, visual disturbances, and contact dermatitis [65]. In this regard, Sebastian et al. developed a highly sensitive and reliable electrochemical sensor by embedding ternary metal oxide comprising ZnO, CuO, and MgO in β-CD functionalized carbon black [66]. Furthermore, the prepared electrochemical sensor was applied for the detection of TBHQ in edible oil samples. Vanillin is one of the most used seasonings in the food industry. However, the extraction process of natural vanillin is complicated, and the high cost can only meet less than 1% of the market demand. Therefore, most of the vanillin used is synthesized by chemical methods [67]. Based on the selective host–guest interaction between vanillin and β-cyclodextrin, Durán et al. developed an optical sensor for vanillin in food samples using CdSe/ZnS quantum dots (QDs) modified with β-CD [68]. It was found that the interaction between vanillin and β-CD–CdSe/ZnS-QDs complex led to the decreased fluorescence of β-CD–CdSe/ZnS-QDs. The method was selective for vanillin with a limit of detection of 0.99 μg/mL.

The illicit incorporation of toxic chemicals into food has increased dramatically over the past decade, posing a serious threat to human health. Melamine is one of them, and because of the adverse health effects of melamine incorporated in food and dairy products on children, adults, and animals, there is an urgent need for selective and sensitive sensors to monitor melamine continuously [69,70]. Xavier et al. synthesized highly robust AgNPs using β-CD as a reducing agent for the colorimetric sensing of melamine [71]. The toxic melamine was detected through the colorimetric response, owing to the host–guest inclusion of melamine into the hydrophobic cavities of β-CD-functionalized AgNPs. Furthermore, the fabricated sensor exhibited a practical applicability in the quantification of melamine in pasteurized milk samples. In another work, Liao et al. constructed an “ON–OFF–ON” sensor for melamine detection based on β-CD-modified carbon nanoparticles (β-CD-CNPs). The sensor was switched “OFF” when Fe^3+^ interacted with β-CD-CNPs and switched “ON” when melamine replaced Fe^3+^. The detection limit was 6.82 ng/mL, which can meet the requirements of actual sample detection [72]. To overcome the tedious procedures for melamine detection, Singh et al. reported the recognition of melamine through the formation of a serendipitously discovered unique supramolecular assembly [73]. The assembly consisted of sulfated cyclodextrin, silver ion, and melamine, which, in turn, encapsulated a molecular rotor dye, Thioflavin-T, to yield a fluorescence turn-on response towards melamine. Overall, this sensor achieved simple, sensitive, and selective detection of melamine.

### 5.2. Foodborne Pathogens

Foodborne pathogens can contaminate the food we consume, causing major foodborne illnesses and even considerable morbidity and mortality [74,75]. Early detection of foodborne pathogens is necessary to prevent foodborne disease outbreaks. The host–guest recognition ability of CDs and their internal hydrophobic and external hydrophilic properties can make it easy to encapsulate various small molecular substances, indicating that they can be used as a platform for developing novel nanosensors [76].

*Listeria* is a foodborne pathogen that may cause infectious listeriosis, leading to meningoencephalitis and sepsis [77]. Li et al. used β-CD and milk protein-coated activated carbon to remove polymerase inhibitors from green leafy matrices and facilitated the recovery of *Listeria monocytogenes*, which could then be coupled with a polymerase chain reaction. This method is fast, sensitive, and specific and can detect and monitor target pathogens in green leafy vegetables, reducing the potential public health risk of *Listeria monocytogenes* [78]. *Staphylococcus aureus* (*S.* aureus) is one of the most prevalent bacteria to transmissible the infectious diseases in hospitals and health care facilities. It causes various diseases in humans, from mild skin infections to fatal sepsis that leads to organ failure in multiple cases [79]. ß-CD has been widely employed as a modifier in voltammetry for the detection of various analytes by preconcentrating the species onto the electrode and thus enhancing its sensitivity for target molecules [80,81]. Gill et al. developed an amperometric electrochemical sensor for the detection and inactivation of methicillin-resistant *S. aureus* (MRSA) [82]. This sensor was based on copper/copper oxide nanocomposites stabilized by β-CD and graphene oxide sheets. Differential pulse voltammetry was used to monitor bacterial capture at the electrode. The sensor achieved a very low limit of detection of 5 CFU/mL and a linear range of 10−10^7^ CFU/mL. 

### 5.3. Pesticides

In modern agricultural production, various pesticides and fungicides are of great significance in improving the quality and yield of crops. It is estimated that the agricultural sector worldwide uses 1–2.5 million tons of pesticides annually to protect crops from pests and diseases [83]. However, the residue of the pesticides has become a global problem. Most pesticides are sprayed on the surface of crops, fruits, and vegetables, contaminating surface water and penetrating through the soil to lower layers of soil and groundwater [84]. Therefore, they can diffuse into the human body through the food chain and eventually accumulate in the body, causing damage to human nerves and organs [85]. Food pesticide residues are a potential threat to humans, so it is necessary to detect pesticide residues in food samples.

Residues of organophosphorus pesticides (OPs) have caused organ dysfunction, cancer, and fetal malformations, even at low concentrations. Methyl parathion (MP) is a specific organophosphorus pesticide that quickly accumulates in food and water because of its bioaccumulation and low solubility [86,87]. Li et al. fabricated an electrochemical sensor for MP detection using β-CD functionalized porous carbon spheres [88]. Porous carbon spheres derived from expired sugarcane juice displayed excellent electrical conductivity, strong adsorption property, and high specific surface area, while β-CD with molecular recognition property promoted the recognition and adsorption of MP. Thanks to the synergistic combination of porous carbon spheres and β-CD, this sensor achieved a wide detection range of 0.01–10 µM for MP analysis. Chadha et al. synthesized noble metal nanoprobes, γ-CD-Ag and γ-CD-Au, by a simple one-step green synthesis using γ-CD as a platform for the colorimetric and Raman sensing of chlorpyrifos [89]. The interaction between the metal particles and chlorpyrifos resulted in colorimetric changes as well as enhancement in the Raman intensity. The γ-CD-Ag and γ-CD-Au nanoprobes showed excellent colorimetric and Raman response toward the pesticide. In another work, Wang et al. established a new non-enzymatic method for the rapid detection of malathion in water [90]. Malathion can block the fluorescence resonance energy transfer (FRET) between the chemical fluorescent probe (donor) and β-CD-coated AgNPs (receptor). Under the optimized conditions, the linear range was from 0.1 to 25 μg/mL.

Organochlorine pesticides (OCPs) are widely used to control plant diseases and insect pests [91]. Although countries have strict controls on the residues of organochlorine pesticides in food, they have not completely banned their use [92,93], and residual organochlorine pesticides can still be found in crops. Sun et al. prepared CD-MOF/TiO_2_ by grafting metal-organic frameworks with CDs as the organic linker onto TiO_2_. A simple and efficient method for OCP detection was established based on dispersive solid-phase extraction and gas chromatography-tandem mass spectrometry (GC-MS/MS) [94]. Zhao et al. fabricated a reduced graphene oxide/cyclodextrin modified GCE (rGO/CD/GCE) for the sensitive electrochemical detection of imidacloprid (IDP). The electrochemical behavior of three different sizes of CDs (α-, β-, γ-CD) functionalized electrodes for IDP was investigated, and the results showed that the supramolecular recognition performance of α-CD composite for IDP was superior to that of other composites [95].

Triazole and chlorothalonil fungicides are commonly used in cultivation, with efficient bactericidal activity [96]. Jing et al. established a CD dispersion liquid–liquid microextraction method, which has been successfully applied to the detection of triazoles and fungicides in water, fruit juice, and vinegar [97]. Sebastian et al. reported the sensitive detection of a novel bifunctional nanocomposite based on carbon nanofibers (CNF), β-CD, and hematite (α-Fe_2_O_3_) nanoparticles for the toxic fungicide carbendazim (CBZ) and the degradation. Carbendazim was detected in actual samples, including apples, oranges, and tomatoes, indicating that the sensor has good practical feasibility [98]. Liu et al. prepared a novel electrochemical sensor for CBZ detection based on β-CD-functionalized carbon nanosheets@carbon nanotubes (β-CD/CNS@CNT) (Figure 3). The practicability of the sensor was verified by the satisfactory actual sample analysis [99].

### 5.4. Antibiotics

Antibiotics are primarily secondary metabolites or artificially synthesized analogs produced by bacteria, mold, or other microorganisms. Antibiotics are widely used in livestock and poultry breeding to prevent and treat various foodborne animal diseases. They are also a common feed additive used to increase the growth rate of livestock [100]. However, the residue of antibiotics can lead to public health, environmental and industrial problems.

Norfloxacin (NFX) in milk increases human resistance to the drugs and threatens public health. Qiu et al. proposed a sensitive method for NFX determination based on SERS using β-CD functionalized AgNPs (β-CD-AgNPs) as substrates [101]. Due to the unique steric size and hydrophilicity of β-CD on the surface of AgNPs, NFX can be captured selectively by weak interactions such as hydrogen bonding interaction and electrostatic interaction. The Raman signal of NFX is largely enhanced when anchoring β-CD on the surface of AgNPs due to SERS effect. Under optimal conditions, the detection limit in standard solution and spiked milk were calculated to be 3.214 pM and 5.327 nM. Excessive fluoroquinolone residues can cause serious safety and health problems due to their potential carcinogenicity and antibiotic resistance. Belenguer-Sapina et al. developed an analytical method based on the solid-phase extraction of three veterinary fluoroquinolones (ofloxacin, norfloxacin, and ciprofloxacin) from milk samples followed by quantitative analysis with liquid chromatography coupled to fluorescence detection [102]. The adsorbent used was a novel mesoporous silica modified with analyte-accessible γ-CD, which facilitates separation through selective host–guest interactions between these analytes. The recoveries of the method ranged from 83% to 92% in water and from 60% to 70% in real milk samples.

### 5.5. Heavy Metals

Heavy metals can accumulate in the human body through the food chain, severely threatening human health because of their side effect. The functional groups such as hydroxyl, carboxyl, and amino groups in CD-based adsorbents are suitable for heavy metals’ fast and effective adsorption [103]. Furthermore, CDs can easily accommodate various suitable small molecules to prepare functional CD inclusion complexes because of their internal hydrophobic and external hydrophilic supramolecular structures. Consequently, these CD inclusion complexes can be used to detect heavy metals with high selectivity and sensitivity.

To improve the separation efficacy of heavy metals, Hassan et al. incorporated β-CD with alginate polymer beads and iron oxide nanoparticles for heavy metal movement (Cd, Ni, Cu, Pb) [104]. The experimental conditions, including adsorption time and the adsorbent dose, were performed to understand the sorption mechanism and evaluate removal efficacy. This demonstrated that electrostatic/covalent binding plays a considerable role in removing heavy metals. In another work, Li et al. designed a novel redox-responsive nanocarrier using β-CD modified nanosilica, which could load and release plant hormones, including salicylic acid (SA) [105] (Figure 4). When the SA-loaded nanoparticles cross the plant cell wall, the disulfide bond can be broken to form sulfhydryl groups under the action of reduced glutathione, thus releasing SA. Meanwhile, the resulting thiol groups exhibited strong affinity toward several heavy metals, such as mercury ions, thus playing a role similar to phytochelatins for detoxification. The release of SA in vitro indicated that the release proceeded much faster in glutathione-rich than in glutathione-free environments. The adsorption behaviors of the redox-responsive nanoparticles toward heavy metals, after phytohormones release, were systematically investigated. Moreover, the synergetic effects on sustained release and metal metals adsorption proved that the redox-responsive CD-modified silica is an effective platform in agricultural applications.

Mercury is the most dangerous and representative heavy metal ion, and even if its concentration is deficient, it will cause serious harm to humans and the environment. Aquatic microorganisms convert mercury ions in the environment into organic mercury such as methylmercury (CH_3_HgX), which can accumulate through the food chain and enter the human body to cause many diseases [106,107]. In this regard, Prabu et al. prepared curcumin, β-CD inclusion complex, which can selectively and sensitively sense mercury ions by absorption and fluorescence detection. This method has the advantages of low cost, good selectivity, high sensitivity, and easy biodegradation, and it has been validated in actual water samples [108]. Based on the self-assembly of β-CD/AD (adamantane), Li et al. constructed a multifunctional fluorescent chemosensor with selective “off–on” behavior that can be reused multiple times. Adamantane-modified fluorescein/cyclodextrin-modified Fe_3_O_4_@SiO_2_ clathrate magnetic nanoparticles have a specific green fluorescence enhancement effect on zinc ions with a detection limit of 4.5 × 10^−7^ M [109]. Celebioglu et al. prepared a highly efficient membrane based on electrospun polycyclodextrin (poly-CD) nanofibers for the removal of heavy metals from water [110]. The poly-CD nanofibers were produced by the electrospinning of CD molecules, followed by heat treatment to obtain an insoluble poly-CD nanofibrous membrane. The membrane was used for the removal of heavy metals. The equilibrium sorption capacity of the poly-CD nanofibrous membrane was found to be 4.54 ± 0.063 mg/g for heavy metals. Cu^2+^ is a well-known heavy metal that plays a vital role in many biological processes, and its concentration directly affects human health. Guo et al. used polyamine-modified CDs as recognition elements in the α-hemolysin (αHL) pore for highly sensitive and selective detection of Cu^2+^ and validated the method for realistic environments by analyzing tap water practicality of Cu^2+^ detection in samples [111]. Mohandas uses photoluminescent supramolecular CD-functionalized copper nanoclusters (CDNCs) as chemical sensors to detect Fe^3+^ in an aqueous solution. The synthesized CDNCs exhibited higher selectivity and sensitivity for Fe^3+^ than other cations and anions, making it an effective method for monitoring iron ions in drinking water. Additionally, the authors successfully demonstrated the potential use of CDNCs-Fe^3+^ complexes in paper sensors and cell imaging [112].

### 5.6. Others

Modern food packaging technology can protect food from physical damage, soil contamination, and microbial spoilage, ensuring food safety [113]. However, some chemicals, such as phthalates and polyvinyl chloride, can migrate from packaging materials into food. Therefore, it is necessary to detect these chemicals. Chen et al. prepared a novel CD-fluorinated covalent organic framework (CD-F-COF) resin to enrich seven benzene series and three perfluorooctane sulfonic acids in food samples (Figure 5). This method can enrich the traces of multiple targets in a rapid and high-throughput manner [114]. Belenguer-Sapina et al. prepared an innovative material based on mesoporous silica modified with CDs to extract and detect endocrine-disrupting chemicals from bottled apple juice [115]. The presence of CDs facilitated the selective extraction of targets. Cromwell et al. reported a new method for detecting phthalates in cheese powder using CD-enhanced fluorescence [116]. Based on the subtle changes in the analyte affinity for the fluorophore and the cyclodextrin cavity, this method offered a straightforward sample preparation strategy and a rapid signal readout mode. This method was able to detect 15 phthalate esters with highly analyte-specific responses and at concentrations as low as 0.12 μM.

Polycyclic aromatic hydrocarbons (PAHs) refer to a group of environmental pollutants. Boon et al. used magnetic nanoparticles to immobilize CD to prepare a novel adsorbent detecting PAHs in rice. The CD polymer showed reasonably good linearity, recovery, and reproducibility after testing on commercial rice [117].

## 6. Concluding Remarks and Outlook

CDs have gathered increasing attention because of their intrinsic structural properties. This paper reviews the recent advances in the research of CDs, deepening the understanding of preparation methods and broadening their applications in food safety analysis. The versatility of CDs has catalyzed innovation in the area of food science. As our understanding of CDs deepens, the potential analytical applications of CDs broaden. The future of CDs in food safety applications holds excellent promise, especially in the area of portable or on-site detection, as well as multiplex analysis. Despite considerable progress of CD inclusion complexes in the food safety analysis, this field still faces many challenges, and the prospects are provided below.

(1) Expanding the preparation methods for CD inclusion complexes

Different preparation methods for CD inclusion complexes have been expanded from saturated aqueous solution to freeze–drying and a few others, but the preparation of clathrates is not environmentally friendly. Therefore, new methods should be explored for the replacement of toxic cross-linking agents and solvents to expand the preparation methods of CD inclusion complexes.

(2) Rational design of CDs

Up to now, some types of CDs are not costly to produce. The following studies might be considered to prepare high-performance CDs. First, it is essential to understand the CD inclusion complex formation mechanism. Second, to reveal the structure–activity relationship of the CD inclusion complex, experimental and computational studies should be incorporated.

(3) Bioeffects of CDs

CDs have many applications in food packaging and food production. When used for food analysis, the fates of CDs in the food matrix should be systematically evaluated, including the detection rate, durability, and reuse. Biocompatibility and biodegradability are common concerns of nanomaterials-based applications. With the advancement of CDs, food detection technology related to CDs will become an emerging research hotspot.

(4) Practical applications

Notably, the subsequent practical application of CDs inclusion complex is also indispensable. Therefore, a comprehensive understanding of absorption–distribution–metabolism–excretion is needed.

(5) Miniaturization and automation for point-of-care detection

Although the developed sensors could provide excellent sensitivity in food safety analysis, few of them are widely used in practical applications due to the cost, stability, and speed. The goals of food analysis should focus on selectivity, reproducibility, and stability in complex matrixes and miniaturization of biosensors by technology.

(6) Multiplexed detection

Some contaminants may coexist in food matrices, which may lead to double toxicity and pose a threat to the environment and living organisms. Therefore, developing sensitive and reliable sensors/biosensors for multiplexed analysis is essential.

(7) Smart packaging

Intelligent packaging methods would be beneficial for the packaging of highly perishable foods. Smart packaging technologies have important practical significance in accelerating the informatization and intelligentization of food analysis. Substantial efforts should be made to develop and translate laboratory finding into a commercial product.

Overall, the development of stable and reliable methods for food safety analysis still has a long way to go. The development of materials and methodologies with superior performance is the main trend of food safety in the future.

## Data Availability

Not applicable.

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
