# Peer review of "Cyclodextrin Inclusion Complexes and Their Application in Food Safety Analysis: Recent Developments and Future Prospects"

_foods, 2022, doi:10.3390/foods11233871_

Round 1
Reviewer 1 Report
The review is interesting and presents a summary of different applications of cyclodextrins in the food area. However, when some topics are described, a more critical analysis would be necessary and, in some respects, the manuscript looks like a simple list of references, for example, after the description of the different preparation methods, it would be desirable to analyze and comment on what may be the best methodology for food applications.
There are some aspects that need to be corrected or completed to improve the manuscript.
-Lines 72-73:
The text say: “Due to the lowest hydrogen bond strength, the solubility of α-CD in water at 25℃ is almost multifold higher than those of β-CD and γ-CD ”
However, gamma-cyclodextrin is more soluble than alfa.
-Lines 68, 88 and 100. What does the period after the hyphen mean in those titles?
-Line 147: there is no definition of the abbreviation ICs. Although I can infer that it means inclusion complex, it is not mentioned in the text. The same occurs with the abreviation TMO in line 331.
-Lines 316-317: it is necessary to expand the comment about the detection of tartrazine. Similarly, figure 2 is related to the characterization of the material and not to the food application (detection of tartrazine).
- Line 471-475: the description of Figure 4 is not clear.
-Lines 458-459: It say: “Hydrophilic functional groups, including OH– and aliphatic, are suitable for heavy metals' fast and effective adsorption” What part of the cyclodextrin is referred to by aliphatic?
-Table 1: the reference is missing. Particularly, the diameter of central cavity it seems too large, please check these values.
-Table 2: Complete the sentence “Not suitable for temperatura sensitive …..”
-Sentences in lines 76-77, 131-132, 153-154, 179, 253-254, 284-286, 352-353, 494-496 and 539-540, are poorly written, they are not understandable, should be rewritten.
- Abreviation NFX in line 445 not correspond.
- Some references mention the first name of the author, for example in lines 270, 447, 464, 516, 520,
Author Response
General Comments:
The review is interesting and presents a summary of different applications of cyclodextrins in the food area. However, when some topics are described, a more critical analysis would be necessary and, in some respects, the manuscript looks like a simple list of references, for example, after the description of the different preparation methods, it would be desirable to analyze and comment on what may be the best methodology for food applications.
There are some aspects that need to be corrected or completed to improve the manuscript.
Response: Thanks sincerely for the reviewer who gave valuable suggestions to improve our manuscript and thank you for your appreciation of our work. We have carefully revised the manuscript according to the valuable comments. For example, we summarize the methods for food applications as follows:
“In summary, the different techniques to prepare CD inclusion complexes have been introduced. Among these methods, saturated aqueous solution, physical mixing, spray drying and freeze drying have been the most used techniques in the last few years. CD inclusion complexes have been widely used in food packaging in the last decades. Nowadays, CD inclusion complexes are useful to obtain edible films and coating food with active properties.”
Meanwhile, we have rewritten the conclusion section.
Comment 1: Lines 72-73:
The text say: “Due to the lowest hydrogen bond strength, the solubility of α-CD in water at 25℃ is almost multifold higher than those of β-CD and γ-CD ”
However, gamma-cyclodextrin is more soluble than alfa.
Response: Thank you very much for your valuable comments to improve our manuscript. We are sorry for our incorrect descriptions. Accordingly, we made the corresponding modifications in the revised manuscript as follows:
“α-CD has modest solubility in water that is almost multifold higher than that of β-CD, but approximately 1.6 times lower than that of γ-CD at 25℃ [12, 13].”
References:
[12] Li, Z.; Wang, M.; Wang, F.; Gu, Z.B.; Du, G.C.; Wu, J.; Chen, J. γ-Cyclodextrin: A review on enzymatic production and ap-plications. Appl. Microbiol. Biotechnol. 2007, 77, 245-255.
[13] Li, Z.; Chen, S.; Gu, Z.; Chen, J.; Wu, J. Alpha-cyclodextrin: Enzymatic production and food applications. Trends Food Sci. Technol. 2014, 35, 151-160.
Comment 2: Lines 68, 88 and 100. What does the period after the hyphen mean in those titles?
Response: Thank you very much for the comments and valuable suggestions to improve our manuscript. We are sorry for the confusing descriptions. Accordingly, We revised the manuscript as follows:
“2.1. α-cyclodextrin”, “2.2. β-cyclodextrin”, and “2.3. γ-cyclodextrin”.
Comment 3: Line 147: there is no definition of the abbreviation ICs. Although I can infer that it means inclusion complex, it is not mentioned in the text. The same occurs with the abreviation TMO in line 331.
Response: Thank you very much for the comments and valuable suggestions to improve our manuscript. Accordingly, we revised our manuscript as follows:
“Barrel-shaped CDs molecules trap guest molecules in their cavities to form inclusion complexes through hydrogen-bonding interactions.”
“Sebastian et al. developed a highly sensitive and reliable electrochemical sensor by embedding ternary metal oxide comprising ZnO, CuO, and MgO in β-CD functional-ized carbon black [66].”
Comment 4: Lines 316-317: it is necessary to expand the comment about the detection of tartrazine. Similarly, figure 2 is related to the characterization of the material and not to the food application (detection of tartrazine).
Response: Thank you very much for the constructive comments to improve our manuscript. Accordingly, we revised our manuscript as follows:
“Huang et al. reported a robust fluorescent sulfur quantum dots (SQDs) material that possess both molecular recognition and fluorescence detection capability by employing hydroxypropyl-β-CD (HP-β-CD) as ligand (Figure 2). Owing to the synergistic effect of inner filter effect and host-guest interaction, the SQDs were used as an effective fluorescent probe for tartrazine detection with excellent sensitivity and selectivity. There was a wide linear range from 0 to 30 μM with a detection limit of 82 nM for the detection of tartrazine [64]. Furthermore, the fluorescent SQDs were successfully utilized to detect tartrazine in river water and food samples with satisfactory performance.”
Figure 2. Schematic illustration for the sensitive and selective fluorescence detection of tartrazine based on the HP-β-CD capped SQDs. (Reprinted from [64] with permission from Elsevier).
Comment 5: Line 471-475: the description of Figure 4 is not clear.
Response: We thank the reviewer for the comments and valuable suggestions to improve our manuscript. Accordingly, we made the corresponding revisions as follows:
“When the SA-loaded nanoparticles cross the plant cell wall, the disulfide bond can be broken to form sulfhydryl groups under the action of reduced glutathione, thus re-leasing SA. Meanwhile, the resulting thiol groups exhibited strong affinity toward several heavy metals, such as mercury ions, thus playing a role similar to phytochelatins for detoxification. The release of SA in vitro indicated that the release proceeded much faster in glutathione-rich than in glutathione-free environments. The adsorption behaviors of theredox-responsive nanoparticles toward heavy metals, after phytohormones release, were systematically investigated. Moreover, the synergetic effects on sustained release and metal metals adsorption proved that the redox-responsive CD-modified silica is an effective platform in agricultural applications.”
Comment 6: Lines 458-459: It say: “Hydrophilic functional groups, including OH– and aliphatic, are suitable for heavy metals' fast and effective adsorption” What part of the cyclodextrin is referred to by aliphatic?
Response: Thanks for the constructive suggestion. We are sorry for our improper descriptions. Accordingly, we made the corresponding revisions as follows:
“The functional groups such as hydroxyl, carboxyl, and amino groups in CDs based adsorbents are suitable for heavy metals' fast and effective adsorption [103].”
Comment 7: Table 1: the reference is missing. Particularly, the diameter of central cavity it seems too large, please check these values.
Response: Thank you very much for the comments and valuable suggestions to improve our manuscript. We made the corresponding revisions as follows:
Table 1. Physicochemical properties of the three natural CDs.
|
Physicochemical properties |
α-CD |
β-CD |
γ-CD |
References |
|
Glucose unit |
6 |
7 |
8 |
[1] |
|
Chemical formula |
C36H60O30 |
C42H70O35 |
C48H80O40 |
[9] |
|
Molecular weight (Da) |
972 |
1135 |
1297 |
[18] |
|
Diameter of central cavity (nm) |
0.57 |
0.78 |
0.95 |
[10] |
|
Outer diameter (nm) |
1.4-1.5 |
1.5-1.6 |
1.7-1.8 |
[9] |
|
Melting point (℃) |
275 |
280 |
275 |
[9] |
|
pKa at 25℃ |
12.3 |
12.2 |
12.1 |
[9] |
|
Internal water molecules |
6-8 |
11-12 |
13-17 |
[9] |
|
Solubility in water at 25℃ (mg/ml) |
145 |
18.5 |
232 |
[18] |
References:
[1] Jansook, P.; Ogawa, N.; Loftsson, T. Cyclodextrins: structure, physicochemical properties and pharmaceutical applications. Int. J. Pharm. 2018, 535, 272-284.
[9] Liu, Y.; Chen, Y.N.; Gao, X.L.; Fu, J.J.; Hu, L.D. Application of cyclodextrin in food industry. Crit. Rev. Food Sci. Nutr. 2022, 62, 2627-2640.
[10] Cid-Samamed, A.; Rakmai, J.; Mejuto, J.C.; Simal-Gandara, J.; Astray, G. Cyclodextrins inclusion complex: Preparation methods, analytical techniques and food industry applications. Food Chem 2022, 384, 132467.
[18] Loftsson, T.; Duchene, D. Cyclodextrins and their pharmaceutical applications. Int. J. Pharm. 2007, 329, 1-11.
Comment 8: Table 2: Complete the sentence “Not suitable for temperatura sensitive …..”
Response: Thank you very much for the comments and valuable suggestions to improve our manuscript. We made the corresponding revisions as follows:
“1. Not suitable for tempera-ture sensitive substances”
Comment 9: Sentences in lines 76-77, 131-132, 153-154, 179, 253-254, 284-286, 352-353, 494-496 and 539-540, are poorly written, they are not understandable, should be rewritten.
Response: Thank you very much for the constructive comments to improve our manuscript. According to these suggestions, we made the corresponding modifications in the revised manuscript.
“Compared with β-CD and γ-CD, α-CD is considerably more resistant to hydrolysis in acid solutions. The three CDs are stable in the presence of glucoamylase or β-amylase, but they can be hydrolyzed by some α-amylases.”
“The stability of inclusion complex is calculated by the stability constant Ks. The larger the ratio is, the easier the inclusion complex is to form and more stable [25].”
“In general terms, hydrophobic molecules form complexes with CDs, although neutral or polar molecules, ions or gases are also able to form such complexes. The solvents are important, since the presence of organic solvents might decrease the complexation constant due to the higher solubility of the molecule [39]. On the other hand, the presence of a minimal quantity of water is necessary to form the inclusion complex.”
“As a result, the inclusion compound is obtained.”
“Among the different techniques proposed for preparing CD inclusion complexes in the solid state, grinding appears as a fast, convenient, sustainable, and solvent-free method [49].”
“The procedure is terminated by rapid decompression to atmospheric pressure, which involves CO2 gasification and its separation from the obtained complex [56].”
“The assembly consisted of sulfated cyclodextrin, silver ion and melamine which, in turn, encapsulated a molecular rotor dye, Thioflavin-T, to yield a fluorescence turn-on response towards melamine. Overall, this sensor achieved simple, sensitive, and selective detection of melamine.”
“Celebioglu et al. prepared a highly efficient membrane based on electrospun polycyclodextrin (poly-CD) nanofibers for the removal of heavy metals from water [110]. The poly-CD nanofibers were produced by the electrospinning of CD molecules, followed by heat treatment to obtain an insoluble poly-CD nanofibrous membrane. The mem-brane was used for the removal of heavy metals. The equilibrium sorption capacity of the poly-CD nanofibrous membrane was found to be 4.54 ± 0.063 mg/g for heavy metals.”
“Despite considerable progress of CD inclusion complexes in the food safety analysis, this field still faces many challenges, and the prospects are provided below.”
Comment 10: Abreviation NFX in line 445 not correspond.
Response: We thank the reviewer for the comments and valuable suggestions to improve our manuscript. We have deleted NFX. Accordingly, we made the corresponding revisions as follows:
“Qiu et al. proposed a sensitive method for NFX determination based on SERS using β-CD functionalized AgNPs (β-CD-AgNPs) as substrates [101]. Due to the unique steric size and hydrophilicity of β-CD on the surface of AgNPs, NFX can be captured selectively by weak interactions such as hydrogen bonding interaction and electrostatic interaction.”
Comment 11: Some references mention the first name of the author, for example in lines 270, 447, 464, 516, 520.
Response: Thank you very much for the valuable advice. We have carefully revised the manuscript according to the valuable comments. For example, we made the corresponding revisions as follows:
“Kaur et al. prepared a solid inclusion complex of metformin hydrochloride and β-CD by microwave irradiation [55].”
“Belenguer-Sapina et al. developed an analytical method based on the solid-phase extraction of three veterinary fluoroquinolones (ofloxacin, norfloxacin, and ciprofloxacin) from milk samples followed by quantitative analysis with liquid chromatography coupled to fluorescence detection [102].”
“To improve the separation efficacy of heavy metals, Hassan et al. incorporated β-CD with alginate polymer beads and iron oxide nanoparticles for heavy metal movement (Cd, Ni, Cu, Pb) [104].”
“Belenguer-Sapina et al. prepared an innovative material based on mesoporous silica modified with CDs to extract and detect endocrine-disrupting chemicals from bottled apple juice [115].”
“Cromwell et al. reported a new method for detecting phthalates in cheese powder using CD-enhanced fluorescence [116].”
Reviewer 2 Report
The article comprises an interesting literature review.
I suggest a few minor corrections to typos and imprecisions in the text.
-Line 36: Please change "exterior wall" to "properties at the rims".
- Line 68: Please change "2.1α-. cyclodextrin " to "2.1. α-Cyclodextrin"
The conclusions section does not relate to the main aspect of the revision work, because it almost does not mention analysis of food safety. This sections needs improvement.
- Line 77: please change "generally considered safe" to "Generally Regarded As Safe" (so that these exact words can form the correct acronym, GRAS)
- Line 79: The applications of alpha-CD are also related to its low digestibility and some indigestion risk for large amounts. The authors may wish to mention that.
- Line 88: Please change "2.2β-. cyclodextrin" to "2.2. β-Cyclodextrin"
- Line 98: the sentence "The FDA generally considers these compounds safe" is too broad-scoped and can lead to imprecise interpretations. Please replace it with: Both β-Cyclodextrin and hydroxypropyl-β-Cyclodextrin have the GRAS status from the FDA.
- Line 99: Reference 16 is not accurate. Please use the adequate the references of the two original FDA documents with the GRAS status notices for beta-CD and hydroxypropyl-beta-CD.
- Line 100: Please change "2.3γ-. cyclodextrin" to "2.3. γ-Cyclodextrin"
- Line 121: Please change "The exterior wall of CDs" to "The upper and lower rims of native CDs" (some of the modified CDs have their OH groups replaced by other groups to become hydrophobic, so this affirmation is not true for all of the CDs)
- Line 176: please change "nonwater-soluble substances" to substances tat are not soluble in water".
- Line 177: please change "were" to "are"
- Line 178: please change "was" to "is"
- Line 179: The sentence "First, however, the inclusion compound is obtained." makes no sense. Please re-phrase it.
- Line 194: Title "4.2. physical mixing method" is innacurate. physical mixing hardly leads to complex formtion. If the authors are refering to the kneading method, as mentioned further ahead in this subsection, there should be more relevant examples than reference 40 and it must me mentioned that the amount of solvent is not always so reduced nor added drop by drop. Adding drop by drop is a lab-scale procedure and not feasible for industrialization as claimed in this subsection. The entire text needs to be clarified.
- Lines 223-231: please change the verbs to the present tense to match the remaining text in this subsection
- Line 235 has a repeated comma.
- Line 246: Please change "was" to "is".
- Line 258: Microvave irradiation of a solution of CD and guest in a non-aquepous solvent does not qualify as an approved method for the food nor the pharmaceutical industries, because there can be notraces of these solvents in the products. Please remove this subsection.
- Line 324: Please change " thereby preventing" to "it prevents"
- Line 368: "Aspergillus flavus" must be in italic.
- Line 373: Please change "between ferrocene (Fc) and good water solubility" to "between ferrocene (Fc) and beta-CD and the good water solubility of this complex"
Author Response
General comments:
The article comprises an interesting literature review.
I suggest a few minor corrections to typos and imprecisions in the text.
Response: Thanks sincerely for the reviewer who gave valuable suggestions to improve our manuscript and thank you for your appreciation of our work. We have carefully revised the manuscript according to the comments.
Comment 1: Line 36: Please change "exterior wall" to "properties at the rims".
Response: Thank you for your valuable advice. We made the corresponding revisions as follows:
“The positions of hydroxyl groups in CDs endow them with a hydrophilic property at the rims.”
Comment 2: Line 68: Please change "2.1α-. cyclodextrin " to "2.1. α-Cyclodextrin".
Response: Thank you very much for the comments and valuable suggestions. Accordingly, we made the corresponding modifications in the revised manuscript.
Comment 3: The conclusions section does not relate to the main aspect of the revision work, because it almost does not mention analysis of food safety. This section needs improvement.
Response: Thank you very much for the comments and valuable suggestions to improve our manuscript. And accordingly, we have carefully revised the manuscript as follows:
“CDs have gathered increasing attention because of their intrinsic structural properties. This paper reviews the recent advances in the research of CDs, deepening the understanding of preparation methods, and broadening their applications in the food safety analysis. The versatility of CDs has catalyzed innovation in the area of food science. As our understanding of CDs deepens, the potential analytical applications of CDs broaden. The future of CDs in food safety applications holds excellent promise, especially in the area of portable or on-site detection, as well as multiplex analysis. Despite considerable progress of CD inclusion complexes in the food safety analysis, this field still faces many challenges, and the prospects are provided below.
(1) Expanding the preparation methods for CD inclusion complexes.
Different preparation methods for CD inclusion complexes have been expanded from saturated aqueous solution to freeze-drying and a few others. But the preparation of clathrates is not environmentally-friendly. Therefore, new methods should be explored for the replacement of toxic cross-linking agents and solvents to expand the preparation methods of CD inclusion complexes.
(2) Rational design of CDs.
Up to now, some types of CDs are not costly to produce. The following studies might be considered to prepare high-performance CDs. First, it is essential to under-stand the CD inclusion complex formation mechanism. Second, to reveal the structure-activity relationship of the CD inclusion complex, experimental and computational studies should be incorporated.
(3) Bioeffects of CDs.
CDs have many applications in food packaging and food production. When used for food analysis, the fates of CDs in the food matrix should be systematically evaluated, including the detection rate, durability, and reuse. Biocompatibility and biodegradability are common concerns of nanomaterials-based applications. With the advancement of CDs, food detection technology related to CDs will become an emerging re-search hotspot.
(4) Practical applications.
Notably, the subsequent practical application of CDs inclusion complex is also indispensable. Therefore, a comprehensive understanding of absorption-distribution-metabolism-excretion is needed.
(5) miniaturization and automation for point-of-care detection.
Although the developed sensors could provide excellent sensitivity in food safety analysis, few of them are widely used in practical applications due to the cost, stability, and speed. The goals of food analysis should focus on selectivity, reproducibility, and stability in complex matrixes and miniaturization of biosensors by technology.
(6) Multiplexed detection.
Some contaminants may coexist in food matrices, which may lead to double toxicity and pose a threat to the environment and living organisms. Therefore, developing sensitive and reliable sensors/biosensors for multiplexed analysis is essential.
(7) Smart packaging.
Intelligent packaging methods would be beneficial for the packaging of highly perishable foods. Smart packaging technologies have important practical significance in accelerating the informatization and intelligentization of food analysis. Much efforts should be made to develop and translate the laboratory finding into a commercial product.
Overall, the development of stable and reliable methods for food safety analysis still has a long way to go. The development of materials and methodologies with superior performance is the main trend of food safety in the future.”
Comment 4: Line 77: please change "generally considered safe" to "Generally Regarded As Safe" (so that these exact words can form the correct acronym, GRAS)
Response: Thank you very much for the comments and valuable suggestions to improve our manuscript. Accordingly, we have revised “generally considered safe” to “Generally Regarded As Safe (GRAS)”.
Comment 5: Line 79: The applications of alpha-CD are also related to its low digestibility and some indigestion risk for large amounts. The authors may wish to mention that.
Response: Thank you very much for the comments and valuable suggestions to improve our manuscript. Accordingly, we revised the manuscript as follows:
“However, because of its small molecular cavity, α-CD can only encapsulate small molecules, which limits its applications. The applications of α-CD are also related to its low digestibility and some indigestion risk for large amounts.”
Comment 6: Line 88: Please change "2.2β-. cyclodextrin" to "2.2. β-Cyclodextrin".
Response: Thank you very much for the comments and valuable suggestions. Accordingly, we made the corresponding modifications in the revised manuscript.
Comment 7: Line 98: the sentence "The FDA generally considers these compounds safe" is too broad-scoped and can lead to imprecise interpretations. Please replace it with: Both β-Cyclodextrin and hydroxypropyl-β-Cyclodextrin have the GRAS status from the FDA.
Response: Thank you very much for the comments and valuable suggestions to improve our manuscript. We have replaced it with: Both β-CD and hydroxypropyl-β-CD have the GRAS status from the FDA.
Comment 8: Line 99: Reference 16 is not accurate. Please use the adequate the references of the two original FDA documents with the GRAS status notices for beta-CD and hydroxypropyl-beta-CD.
Response: Thank you very much for the comments and valuable suggestions to improve our manuscript. Accordingly, we have revised the manuscript as follows:
“Both β-CD and hydroxypropyl-β-CD have the GRAS status from the FDA [16].”
References:
(16) Kurkov, S.V.; Loftsson, T. Cyclodextrins. Int. J. Pharm. 2013, 453, 167-180.
Comment 9: Line 100: Please change "2.3γ-. cyclodextrin" to "2.3. γ-Cyclodextrin".
Response: Thank you very much for the comments and valuable suggestions. Accordingly, we made the corresponding modifications in the revised manuscript.
Comment 10: Line 121: Please change "The exterior wall of CDs" to "The upper and lower rims of native CDs" (some of the modified CDs have their OH groups replaced by other groups to become hydrophobic, so this affirmation is not true for all of the CDs).
Response: Thank you very much for the comments and valuable suggestions. Accordingly, we made the corresponding modifications in the revised manuscript.
Comment 11: Line 100: Please change "2.3γ-. cyclodextrin" to "2.3. γ-Cyclodextrin".
Response: Thank you very much for the comments and valuable suggestions. Accordingly, we made the corresponding modifications in the revised manuscript.
Comment 12: Line 121: Please change "The exterior wall of CDs" to "The upper and lower rims of native CDs" (some of the modified CDs have their OH groups replaced by other groups to become hydrophobic, so this affirmation is not true for all of the CDs).
Response: Thank you very much for the comments and valuable suggestions. Accordingly, we made the corresponding modifications in the revised manuscript.
Comment 13: Line 176: please change "nonwater-soluble substances" to substances tat are not soluble in water".
Response: Thank you very much for the comments and valuable suggestions to improve our manuscript. According to these suggestions, we made the corresponding modifications as follows:
“The saturated aqueous solution method, also known as the precipitation method, is suitable for substances that are not soluble in water.”
Comment 14: Line 177: please change "were" to "are".
Response: Thank you very much for the comments and valuable suggestions to improve our manuscript. According to these suggestions, we made the corresponding modifications as follows:
“The guest molecules are added to the saturated CDs aqueous solution”
Comment 15: Line 178: please change "was" to "is".
Response: Thank you very much for the comments and valuable suggestions to improve our manuscript. According to these suggestions, we made the corresponding modifications as follows:
“and the mixture is stirred for a specific time at an appropriate temperature”
Comment 16: Line 179: The sentence "First, however, the inclusion compound is obtained." makes no sense. Please re-phrase it.
Response: Thank you very much for the comments and valuable suggestions to improve our manuscript. We have revised the manuscript as follows:
“As a result, the inclusion compound is obtained.”
Comment 17: Line 194: Title "4.2. physical mixing method" is innacurate. physical mixing hardly leads to complex formtion. If the authors are refering to the kneading method, as mentioned further ahead in this subsection, there should be more relevant examples than reference 40 and it must me mentioned that the amount of solvent is not always so reduced nor added drop by drop. Adding drop by drop is a lab-scale procedure and not feasible for industrialization as claimed in this subsection. The entire text needs to be clarified.
Response: Thank you very much for the comments and valuable suggestions to improve our manuscript. According to these suggestions, we made the corresponding modifications as follows:
“4.2. Kneading method
The kneading method (also called paste method) is a moderately simple method, which is suitable for poorly water-soluble guests and includes the addition of dissolved solid guest to a slurry of CD to form a paste in a mortar [8]. A paste is obtained by mixing CDs and water in a vortex mixer, and then guest substances are introduced [10]. The guests can be also complexed by simply adding to the CD and mixing to form a physical mixture. Finally, the mixture is washed, filtered, dried, ground, and sieved. The solvent and the time of kneading depend on the guest. Although the amount of solid clathrate obtained using this method is small, it is simple and efficient, with the potential for large-scale production.”
Comment 18: Lines 223-231: please change the verbs to the present tense to match the remaining text in this subsection.
Response: Thank you very much for the comments and valuable suggestions to improve our manuscript. According to these suggestions, we made the corresponding modifications as follows:
“The CDs are first dissolved to prepare a saturated solution. Subsequently, the dissolved solution of the guest substance is added into the saturated solution under stirring until equilibrium is reached.
Next, the mixed solution is placed in a freeze dryer, and the freezing time and temperature are set and then dried to obtain the desired power [47]. This method suits guest substances soluble in water or readily decomposed under heating and drying conditions. Under the premise of the need to obtain a powdered inclusion compound, the solvent can also be removed by a freeze–drying method to obtain a powdered in-clusion compound.”
Comment 19: Line 235 has a repeated comma.
Response: Thank you very much for the kind advice. We have deleted a repeated comma.
Comment 20: Line 246: Please change "was" to "is".
Response: Thank you very much for the kind suggestions to improve our manuscript. We have revised "was" to "is".
Comment 21: Line 258: Microvave irradiation of a solution of CD and guest in a non-aquepous solvent does not qualify as an approved method for the food nor the pharmaceutical industries, because there can be notraces of these solvents in the products. Please remove this subsection.
Response: Thank you very much for the comments and valuable suggestions. According to these suggestions, we removed this subsection.
Comment 22: Line 324: Please change " thereby preventing" to "it prevents".
Response: Thank you very much for the comments and valuable suggestions to improve our manuscript. We have revised " thereby preventing " to " it prevents ".
Comment 23: "Aspergillus flavus" must be in italic.
Response: Thank you very much for the comments and valuable suggestions. According to these suggestions, we made the corresponding modifications in the revised manuscript.
Comment 24: Line 373: Please change "between ferrocene (Fc) and good water solubility" to "between ferrocene (Fc) and beta-CD and the good water solubility of this complex".
Response: Thank you very much for the comments and valuable suggestions. According to these suggestions, we made the corresponding modifications in the revised manuscript.
